Burden changes in notifiable infectious diseases in Taiwan during the COVID-19 pandemic

http://orcid.org/0000-0003-0304-1628 Yang Ying-Fei 1
Chen Yu-Miao 2
Chen Si-Yu 2
Chiu Po-Hao 2
http://orcid.org/0000-0002-3998-0355 Chen Szu-Chieh 2 3 scchen@csmu.edu.tw
1 Department of Bioenvironmental Systems Engineering, National Taiwan University , Taipei , Taiwan
2 Department of Public Health, Chung Shan Medical University , Taichung , Taiwan
3 Department of Family and Community Medicine, Chung Shan Medical University Hospital , Taichung , Taiwan
Thorrington Dominic
Electronic publication date: 2024 Sep 9
Publication date: 2024
Volume: 12
Electronic Location ID: e18048
Received 2024 May 9; Accepted 2024 Aug 15
Copyright: © 2024 Yang et al.
Copyright year: 2024
Copyright holder: Yang et al.
License: This is an open access article distributed under the terms of the Creative Commons Attribution License, which permits unrestricted use, distribution, reproduction and adaptation in any medium and for any purpose provided that it is properly attributed. For attribution, the original author(s), title, publication source (PeerJ) and either DOI or URL of the article must be cited.
License URL: https://creativecommons.org/licenses/by/4.0/

Keywords: Disease burden, SARS-CoV-2, Public health, Notifiable infectious diseases (NIDs), Non-pharmaceutical interventions (NPIs)

Funding: Ministry of Education (Taiwan) PMN1120555 This study received support from Ministry of Education (Taiwan) under grant number PMN1120555. The funders had no role in study design, data collection and analysis, decision to publish, or preparation of the manuscript.

==============================
Background

This study aimed to assess the impact of the COVID-19 pandemic on the disease burden of Taiwan’s notifiable infectious diseases (NIDs). We compared disease burdens between the pandemic and pre-pandemic year of 2020 (with non-pharmaceutical interventions (NPIs)) and 2010 (without NPIs), respectively, to understand the overall pandemic impact on NIDs in Taiwan.

Methods

Forty-three national NIDs were analyzed using the Statistics of Communicable Diseases and Surveillance Report by estimating the premature death and disability via different transmission categories, sex, and age groups. The study evaluated the impact of diseases by assessing the years lost due to death (YLLs), the duration of living with disability (YLDs), and the overall disability-adjusted life years (DALYs) by measuring both the severity of the illness and its duration.

Results

Taiwan recorded 1,577 (2010) and 1,260 (2020) DALYs per million population and lost 43 NIDs, decreasing 317 DALYs per million population. Tuberculosis, HIV/AIDS and acute hepatitis B/D were the leading causes of DALYs, accounting for 89% (2010) and 77% (2020).

Conclusion

Overall, this study provided the first insight of changes in disease burdens in NIDs between pre- and post-COVID-19 based on a nationwide viewpoint for further preventive measures and interventions to be focused on specific diseases by associated health administrations and policies.

Introduction

Since March 10, 2023, numerous countries worldwide had huge impacts by the impact of the novel coronavirus disease 2019 (COVID-19), tallying a total of 676,609,955 confirmed cases and 6,881,955 reported deaths (https://coronavirus.jhu.edu/map.html) (Johns Hopkins University, 2023). As of 5 April 2023, Taiwan had felt the repercussions of the pandemic, with 10,241,498 COVID-19 cases and 16,858 deaths reported (https://covid-19.nchc.org.tw/) (Taiwan Centers for Disease Control (Taiwan CDC), 2019). In response to the COVID-19 pandemic, a variety of public health interventions have been implemented by Central Epidemic Command Center (CECC) of Taiwan to mitigate the impact of the pandemic, including both non-pharmaceutical interventions (NPIs) (case identification, border control, quarantine of suspected cases, proactive case finding and public education) (Wang, Ng & Brook, 2020) and pharmaceutical measures (vaccines, antibodies, etc.,) (Knoll & Wonodi, 2021; Suzuki & Ishihara, 2021; Munro et al., 2021).

During the pandemic, numerous investigations were adopted to evaluate the influence of NPIs on influenza and various respiratory pathogens (Baker et al., 2022; Singh et al., 2021; Huang et al., 2021; Lei et al., 2021). Pathogens transmitted through alternative routes, such as through food, water, sexual activity, or vectors, could also be influenced by significant shifts in human social activity and behavior, altering the dynamics of person-to-person contact (Lucchini et al., 2021). So far, the effect of NPIs on the epidemiology of different modes of infectious diseases were estimated in various countries, including Germany (Ullrich et al., 2021), Netherlands (van Deursen et al., 2022; van den Wijngaard et al., 2015), China (Geng et al., 2021; Zhang et al., 2023), Japan (Hibiya et al., 2022), and Taiwan (Hung et al., 2022). In Germany, national surveillance data were scrutinized to calculate the relative fluctuation in case numbers from week 2020-10 to 2020-32 (during the pandemic and implementation of NPIs), juxtaposed with the period from week 2016-01 to 2020-09 (Ullrich et al., 2021). A descriptive study in Taiwan also found that the total case count during the pandemic exceeded that of the pre-pandemic period, with 39,430 cases reported during 2021–2022 compared to 38,287 cases during 2018–2019, accounting increment of 24 non-airborne/droplet-transmitted NIDs in Taiwan (Hung et al., 2022).

Understanding the NID changes from pre-pandemic to pandemic in Taiwan could facilitate informing the public and decision-makers of the impacts and economic burdens of COVID-19 in order to alleviate huge burdens on healthcare services by implementing better strategies. The disability-adjusted life years (DALYs) was used to estimate the amount of time, ability, or activity lost by an individual due to disability or death caused by a disease (Murray, 1994). Thus far, studies assessing disease burdens or the direct effects of COVID-19 on population health have been conducted in several countries, such as Germany (Rommel et al., 2021), Denmark (Pires et al., 2022), the USA (Quast et al., 2022), Scotland (Wyper et al., 2022), or involvement of multiple countries (Pifarré I Arolas et al., 2021; Cuschieri et al., 2021). However, limited information regarding the disease burdens caused by notifiable infectious diseases (NIDs) between pre-pandemic and pandemic period in Taiwan could be found in existing literature. Therefore, our primary objective was to quantitatively estimate the overall DALYs to furnish regulatory authorities with additional data for the development of enhanced control measures. This study aimed to (i) employ the Statistics of Communicable Diseases and Surveillance Report (Taiwan Centers for Disease Control (Taiwan CDC), 2021) to analyze the overall changes in NIDs from pre-pandemic (2010; without NPIs) to pandemic (2020; with NPIs) period, and (ii) assess the premature mortality and disability impact by computing the quantities of life-years lost, categorized by disease, gender, and age groups in Taiwan.

Materals and methods

Screening for notifiable infectious diseases

We analyzed epidemiological data on 98 NIDs in 2010 and 2020 from the database of Taiwan Centers for Disease Control (Taiwan CDC). Factors of disease burdens were collected from Abbafati et al. (2020) and published research. The process of determining the inclusion of diseases in this study is shown in Fig. S2. We excluded disease categories without associated cases or deaths (n = 24 NIDs) and those with no surveillance data shown on the Statistics of Communicable Diseases and Surveillance Report (n = 26 NIDs) (Taiwan Centers for Disease Control (Taiwan CDC), 2021). In 2016, the National Health Insurance Administration of the Ministry of Health and Welfare announced that the ICD-10 Clinical Modification (ICD-10-CM) would replace the ICD-9-CM, and the classification scheme of the ICD-10 was more detailed and complicated than that of the ICD-9. However, diseases may be affected by code conversion. For example, the ICD-10 code for Chikungunya fever is A920, but it is classified as tropical hemorrhagic fever in ICD-9 (065.4). In addition, while 440.3 (ICD-10) refers to septicemia is caused by invasive pneumococcal disease, the only code for septicemia is 038 in ICD-9 without further differentiation among disease causes. Therefore, it would be difficult to categorize a single disease, since it could be resulted from two distinct diseases that could not be counted. Moreover, three NIDs were excluded because of a lack of relevant estimates of factors to derive disease burdens. Ultimately, 43 diseases were included in the calculation.

Epidemiological data

We chose 2010 as a pre-pandemic baseline year without non-pharmaceutical interventions (NPIs) to compare with NIDs during the pandemic year of 2020, when NPIs were implemented. To gain a deeper understanding of the disease burden trends before the pandemic, we also analyzed DALYs from 2005 to 2019 in Taiwan, as shown in Fig. S1. The results indicated that the shares of the total disease burden by cause remained steady during this period. Therefore, 2010 was selected as a representative pre-pandemic baseline year to compare with the NIDs and disease burdens in 2020.

The relative change (%) in the number of cases and deaths of NIDs in Taiwan in 2010 and 2020 is shown in Table S1. These diseases could be classified into five categories based on their modes of transmission: (1) airborne/droplet (AD) transmitted, (2) fecal-oral (FO) transmitted, (3) direct-contact (DC) transmitted, (4) vector-borne (VB) transmitted, and (5) sexually transmitted and blood-borne (SB) diseases.

Airborne/droplet (AD) transmitted diseases had the highest number of cases in 2010 and decreased by 59.6% in 2020. Specifically, tuberculosis had the highest number of cases both in 2010 and 2020, with a 40.9% decrease in 2020 when compared to 2010. However, the cases of Legionnaires’ disease increased significantly from 2010 to 2020, with a relative change of 219.6%, especially among individuals aged 40 and older.

Sexually transmitted and blood-borne (SB) diseases had the highest number of cases in 2020, with a 58.6% increase when compared to that of 2010. Notably, the case numbers of acute hepatitis C, gonorrhea, and syphilis increased by 1,368.3%, 212.7%, and 35.7%, respectively, in 2020, when compared to that of 2010. Table S2 displays the codes translated from ICD-9-CM to ICD-10-CM in Taiwan.

Burden of disease

The DALY calculations for confirmed cases and fatalities stemming from disease notification categories under surveillance in Taiwan for 2010 and 2020 were determined using the subsequent formula. Both the years lost due to death (YLLs) and the duration of living with disability (YLDs) were computed separately and then amalgamated into a unified summary measure (Eq. (1)).

(1) DALYs=YLLs+YLDs

The DALYs corresponding to each age-specific category and disease severity were multiplied by the count of NIDs cases in both 2010 and 2020. Subsequently, these DALYs were divided by the respective total population figures for 2010 and 2020 to yield DALY estimates per million population per year.

YLLs were computed by multiplying the number of deaths within a specific age group by the standard life expectancy corresponding to that age group (Eq. (2)).

(2) YLLs=∑x,y⁡(Mx,y×L1x,y),

where Mx,y denotes the age-specific (x) and sex-specific (y) count of NIDs-related deaths. The age categories (x) were less than 1 (<1), 1–4, 5–14, 15–24, 25–39, 40–64, and 65+ years. L1x,y represents the years of life lost due to premature death across different age groups. We employed the life expectancy at the midpoint of each age group, matching the available age-specific mortality data. The 2010 and 2020 life tables were utilized, reflecting life expectancy variations based on birth year (Taiwan Ministry of the Interior (Taiwan MOI), 2010–2020). The life expectancy in 2020 for (<1), 1–4, 5–14, 15–24, 25–39, 40–64, and 65+ age groups were 76.58/83.93 (M/F), 74.43/81.77, 67.51/74.83, 57.72/64.95, 45.71/52.75, 28.18/33.86 and 11.41/14.54 years, respectively (Table S3).

YLDs were calculated by multiplying the number of new cases by the average duration of disability and the disability weightings corresponding to the disease stages (Eq. (3)).

(3) YLDs=∑x,y,z⁡(Ix,y,z×DWz×L2,z),

where Ix,y,z denotes age/sex-specific (x, y) confirmed cases at various disease severities (z); DWz signifies the disability weights (DWs) specific to each disease severity (z), and L2,z indicates the disability duration (in years) specific to each disease severity (z). The DW reflected the severity of the disease and the value between 0 (good health) and 1 (death). The unit of DALY calculation is the person-year (Rommel et al., 2021).

Disease weights and duration of disability

The disease severities (z) were classified as two levels of moderate/severe or three levels of mild/moderate/severe, respectively, based on the available data source in specific diseases. Hence, Appendix Tables S4–S7 list the disease weights and the duration of disability for different NIDs which are classified according to the transmission route. Furthermore, we incorporated disability weights (DWs) for COVID-19. The severity-specific DWs were as follows: asymptomatic, 0; mild, 0.006; moderate, 0.051; severe, 0.133; and very severe/critical, 0.655 (Wyper et al., 2022; Haagsma et al., 2015; World Health Organization (WHO), 2021).

In addition, the disability durations specific to each severity (z) are as follows: 14 days for asymptomatic, mild, and moderate cases; 21 days for severe cases; and 32 days for critical cases (https://www.who.int/docs/default-source/coronaviruse/who-china-joint-mission-on-covid-19-final-report.pdf) (World Health Organization (WHO), 2021). Other estimates to derive disease burdens were adopted from Abbafati et al. (2020). Besides, this study used the average value to calculate if there was a distribution or range of distribution for severity, disease weights, and disability duration.

Results

DALYs per million population

The NIDs were 1,577 and 1,260 DALYs per million population in 2010 and 2020, revealing that the disease burdens were decreased by 317 DALYs per million population in 2020. The leading causes of DALYs were tuberculosis, human immunodeficiency virus (HIV)/acquired immune deficiency syndrome (AIDS), and acute hepatitis B/D, accounting for 89% and 77% in 2010 and 2020, respectively. The top three contributors to YLLs in 2010 were tuberculosis (47%), HIV/AIDS (19%), and acute hepatitis B/D (17%), and were tuberculosis (34%), acute hepatitis B/D (21%), and influenza cases with severe complications (20%) in 2020. Also, the top three contributors to YLDs for 2010 were tuberculosis (83%), HIV/AIDS (12%), and syphilis (3%), and were tuberculosis (78%), HIV/AIDS (14%), and syphilis (7%) in 2020 (Appendix Fig. S3).

Of the five modes of transmission diseases, sexually transmitted and blood-borne (SB) and fecal-oral (FO) transmitted diseases had higher numbers of cases in 2020 than that of 2010 (Figs. 1A and 1B). However, the burden of SB diseases in 2020 became lower than that of 2010 after standardization since the number of HIV/AIDS cases and deaths in 2020 had decreased significantly (Figs. 1C and 1D). In addition, compared with 2010, the percentage of change in DALYs of FO transmitted diseases increased by 51% in 2020, while the rest decreased (Fig. 1E).

Figure 1 (A, B) The number of cases of notifiable infectious diseases (NIDs) sorted by different modes of transmission and (C, D) their associated disability-adjusted life years (DALYs) per million populations in Taiwan in the year of 2010 and 2020. (E) The percentage of change in DALYs per million populations in the year of 2010 and 2020 by different modes of transmission.

The symbols indicate the airborne/droplet transmitted diseases (AD), sexually transmitted and blood-borne diseases (SB), fecal-oral transmitted diseases (FO), direct-contact transmitted diseases (DC), and vector-borne transmitted diseases (VB), respectively.

Total DALYs per million population in 2020 were lower than that of 2010 due to significant reductions in DALYs via AD, SB, DC, and VB transmitted diseases (Figs. 2A and 2B). The relative contributions of YLLs and YLDs to DALYs varied by the modes of transmission category. The disease burdens of YLLs/YLDs were 893/832 per million population and 684/428 per million population in 2010 and 2020, respectively. Overall, in 2010 and 2020, YLLs contributed 57% and 66% towards the DALYs, with the remaining 43% and 34% attributed to YLDs (Figs. 2C and 2D). Regarding sex, the disease burdens of DALYs were 1,154/921 per million population and 423/339 per million population in 2010/2020 for males and females, respectively (Fig. 2E). In addition, the disease burdens of DALYs increased with age. Among all age groups, the population with age ≥65 years old contributed the highest DALYs (689 per million population and 612 per million population, 44% and 49%) in 2010 and 2020, respectively (Fig. 2F).

Figure 2 YLLs and YLDs for five modes of transmission of 43 notifiable infectious diseases (NIDs) in (A) 2010 and (B) 2020. Overall contribution percentage in (C) 2010 and (D) 2020. Burden change of disability-adjusted life years (DALYs) in 2010 and 2020 for specific (E) Gender/Mode of transmission and different. (F) Age groups.

Airborne/droplet-transmitted diseases

Among the 14 airborne/droplet (AB) transmitted diseases, tuberculosis, influenza cases with severe complications, and invasive Haemophilus influenzae Type b infection were the diseases that contributed the most to the disease burden in both 2010 and 2020, with the DALYs (males and females) of 990/612 per million population (in 2010/2020), 62/166, 22/15, respectively; for males, the DALYs were 687/440 per million population (in 2010/2020), 38/105, 17/7, respectively; for females, the DALYs were 303/172 per million population (in 2010/2020), 24/61, 5/8, respectively (Figs. 3A and 3B). For other AB transmitted diseases, the DALYs per million population (in 2010/2020) of severe pneumonia with novel pathogens, multidrug-resistant tuberculosis, mumps, Legionnaires’ disease, Hantavirus syndrome, pertussis, meningococcal meningitis, complicated varicella, Q fever, measles, rubella were 0/8.5, 6.73/3.1, 0.13/1.1, 0.02/0.1, 0/0, 0.02/0, 0.02/0, 0/1.9, 1/0, 0/0, and 0/0, respectively. Interestingly, the DALYs of influenza cases with severe complications increased by 168% in 2020 compared to that of 2010. On the contrary, the DALYs of tuberculosis and invasive Haemophilus influenzae Type b infection decreased by 38% and 32% in 2020. Considering gender, males accounted for approximately 70% of the DALYs of the AB diseases in 2010 and 2020.

Figure 3 DALYs estimation for airborne and droplet transmitted diseases of 14 NIDs in (A) 2020 and (B) 2010. Temporal variation in case numbers notified in Jan. to Dec. in 2010, compared to average cases in 2016–2019 (pre-pandemic) and 2020–2022 (pandemic), of a selection of infectious diseases including (C) Tuberculosis, (D) Influenza cases with severe complication, and (E) Invasive haemophilus influenza Type b infection.

The scale of the secondary vertical axis shows the average monthly cases of COVID-19. The formal classification “Severe Pneumonia with Novel Pathogens” refers to COVID-19.

Figures 3C–3E illustrate the temporal variation in case numbers of tuberculosis, influenza cases with severe complications, and invasive Haemophilus influenzae Type b infection, respectively. The average numbers of the three selected disease cases in 2020–2022 (pandemic) were lower than those in 2016–2019 (pre-pandemic) and 2010.

Sexually transmitted and blood-borne diseases

Regarding the five sexually transmitted and blood-borne (SB) diseases, most of the burdens of disease in both 2010 and 2020 were attributed to HIV/AIDS, acute hepatitis B/D/C, with the DALYs (males and females) of 253/175 per million population (in 2010/2020), 154/174, 44/52, respectively; for males, the DALYs were 240/167 per million population (in 2010/2020), 112/123, 23/27, respectively; for females, the DALYs were 13/8 per million population (in 2010/2020), 42/51, 21/24, respectively (Figs. 4A and 4B). The DALYs of HIV/AIDS decreased by 31% in 2020 compared to that of 2010. In contrast, the DALYs of acute hepatitis B/D/C increased by 13% and 18% in 2020. From a gender perspective, males accounted for approximately 81% of the DALYs of SB diseases in both 2010 and 2020.

Figure 4 DALYs estimation for sexually and blood-borne transmitted diseases of 5 NIDs in (A) 2020 and (B) 2010. Temporal variation in case numbers notified in Jan. to Dec. in 2010, compared to average cases in 2016–2019 (pre-pandemic) and 2020–2022 (pandemic), of a selection of infectious diseases (C) HIV/AIDS, (D) acute hepatitis B/D, and (E) acute hepatitis C.

The scale of the secondary vertical axis shows the average monthly cases of COVID-19.

Figures 4C–4E illustrate the temporal variation in case numbers of HIV/AIDS and acute hepatitis B/D/C, respectively. The average numbers of HIV/AIDS cases were lower in 2020–2022 (pandemic) compared to those in 2010 and 2016–2019 (pre-pandemic), in an ascending order (Fig. 4C). Nevertheless, the average numbers of acute hepatitis C cases in 2020–2022 (pandemic) were higher than those in 2016–2019 (pre-pandemic) and 2010, in a descending order (Fig. 4E).

Discussion

This study aimed to quantitatively estimate the total DALYs to examine the overall changes in NIDs from pre-pandemic in 2010 (without NPIs) to COVID-19 pandemic in 2020 (with NPIs) and assess the premature mortality and disability impact by computing the quantities of life-years lost, categorized by disease, gender, and age groups in Taiwan. Overall, the total estimated DALYs of 1,577 and 1,260 DALYs per million population were lost from NIDs in 2010 and 2020, respectively, with a decrease of 317 DALYs per million population. Lai et al. (2021) also found that the NID cases decreased 2,574 from 2019 to 2020 with a percentage change of −10.5%, which could be ascribed to aggressive COVID-19 control measure implementations since early 2020 in Taiwan. Consistent with our results, it was found that change in incidence of vector-borne diseases were 2.4 (−55.0%) from 2019 to 2020 (Lai et al., 2021). Hung et al. (2022) also evidenced vector-borne NIDs declined during pandemic in Taiwan. Furthermore, we found tuberculosis, human immunodeficiency virus (HIV)/acquired immune deficiency syndrome (AIDS), and acute hepatitis B/D were the leading causes of DALYs, accounting for 89% and 77% in 2010 and 2020, respectively. In accordance with our results, Lai et al. (2021) evidenced the three sexually transmitted diseases (syphilis, HIV, and AIDS) showed lower cases in 2020 than 2019.

COVID-19 is a respiratory infectious disease transmitted primarily through inhalation, direct or indirect contact, mainly via person-to-person contact or exposure to respiratory droplets expelled when an infected individual coughs or sneezes (Taiwan Centers for Disease Control (Taiwan CDC), 2019). Non-pharmaceutical interventions (NPIs), such as wearing a mask, social distancing, or washing hands frequently with soap to prevent the spread of the disease, may reduce the transmission of COVID-19. In the early pandemic period, Taiwan Central Epidemic Command Center was established in the early 2020 to organize associated resources and develop control measures such as implementation of effective policies and interventions (Lai, Lee & Hsueh, 2023). Taiwan Center of Disease Control (TCDC) has implemented different levels of NPIs, including wear facemasks in public transportations, healthcare facilities, indoor public space and physical distancing at restaurants and populated public venues (since February 6th, 2020), hand hygiene, border control (since March 7th, 2020), introduce of digital technology incorporating big data, quarantine of COVID-19 cases (since March 19th, 2020), and travel and gathering restriction (since March 19th, 2020). Taiwanese governments also strictly executed international border control by requesting all arriving passengers a 14-day compulsory quarantine with active surveillance on respiratory symptoms and body temperature (Chen et al., 2021; Lai, Lee & Hsueh, 2023; TCDC, https://www.cdc.gov.tw/Category/ListContent/zKNFqsVWxUoUqE6fyhmBNA?uaid=L-aa5EUo4UIJFsJlT7tAQw).

However, it is challenging to explain the burden changes by multicausality of how the NPIs influenced people’s social lifestyle, sexual behavior, and the transmission dynamics of sexually transmitted and blood-borne diseases. Interpreting the changes in the frequency of sexual contact and the infection status of these diseases in the real world also remains a complex task. For example, changes in the behavior of healthcare seeking, attendance, and utilization could lead to under-diagnosis and under-reporting of NIDs (Ullrich et al., 2021; Hung et al., 2022). Individuals with sexually transmitted and blood-borne diseases or NID infection may be hesitant or reluctant to visit healthcare facilities for screening, diagnosis, or treatment due to the fear of SARS-CoV-2 infection.

Specifically, for airborne and droplet transmitted diseases, influenza case with severe complications increased significantly from 2010 to 2020. However, different to the trend of DALY estimates, the influenza cases with severe complications decreased during pandemic (2020–2022) when compared with that of 2010. Previous studies showed collateral effects of COVID-19 that lower numbers of severe complications from influenza in pandemic period were observed when compared to that of pre-pandemic period (Hsu et al., 2020; Tang, Lai & Chao, 2022). We also found that DALYs of tuberculosis decreased from 2010 to 2020, which is consistent with Lai & Yu (2020) that infection control measures had positively impact on respiratory infectious diseases such as pulmonary tuberculosis. Similarly, the decreased DALYs and case numbers of sexually and blood-borne transmitted diseases such as HIV/AIDS we observed during pandemic is also consistent with Lai & Yu (2020) that the weekly HIV case numbers significantly decreased in 2020 when compared to that of 2018. Our findings revealed variations in the disease burdens of NIDs based on gender, with the male population accounting for a higher proportion of DALYs (73%) compared to the female population (27%). Also, the disease burdens caused by the males were nearly three times higher than those of females. In terms of age, the disease burdens of DALYs increased with age. The population aged ≥65 years old contributed nearly half of DALYs (44% in 2010 and 49% in 2020). However, different to our results, Geng et al. (2021) observed a notable age gap, finding that respiratory diseases and gastrointestinal or enteroviral diseases were more prevalent among children and adolescents in China when compared to adults. But sexually transmitted or bloodborne diseases and vector-borne diseases were found to be higher in adults than children. Also, differences between sexes were relatively small that only slightly higher incidence of gastrointestinal or enteroviral disease was observed in males.

Our study discovered that sexually transmitted and blood-borne diseases, especially HIV/AIDS and acute hepatitis B/D/C are significant contributors to the disease burden among all NIDs in Taiwan. We also found that the DALYs of HIV/AIDS decreased in 2020 compared to 2010, which was due to decreases in cases and deaths. The number of HIV/AIDS cases in Taiwan has consistently declined each year from 2020 to 2022, with a more considerable decrease in 2020 (Taiwan Centers for Disease Control (Taiwan CDC), 2010–2020). However, for the burden of acute hepatitis, the DALYs increased in 2020 compared to that of 2010, which is mainly attributed to the rise of acute hepatitis C cases and deaths from acute hepatitis B. In accordance with our results of increase in case numbers of gonorrhea and syphilis, Lee et al. (2021) reported that the number of gonorrhea cases in Taiwan during the first 8 months of 2020 exceeded the cases reported during the same period between 2015 and 2019.

Limitation and implication of the study

Several limitations of this study warrant mention. Firstly, we did not consider factors that may affect the occurrence of infectious diseases, such as climate change and vaccination rates. Secondly, the estimates of disease burden factors used in the study (disability weight, duration of disease, and severity distribution) were mainly adopted from the Global Burden of Disease (GBD) study and published research due to the limited data sources applicable to Taiwan (Global Burden of Disease Study 2013 Collaborators, 2015). Our methodology is aligned with the concept of World Health Organization (WHO) (2020) approach that the disability weights reflect the general population judgments about the ‘healthfulness’ of defined states without taking any judgments of quality of life or the worth of persons or the social undesirability or stigma of health states. However, although we did consider age/sex-specific confirmed cases into YLD estimations, we couldn’t include factors including age and sex in DW estimates based on the previous studies, due to the difficulties in comprehensively finding disease-specific DWs (Appendix Tables S4–S7). The above reasons may lead to overestimation or underestimation in results such as the disability-adjusted life years.

Overall, this study provided the first insight of changes in disease burdens in NIDs between pre- and post-COVID-19 based on a nationwide viewpoint. Different from the previous study in Taiwan (Lai et al., 2021), we investigated the difference in disease burdens resulting from notifiable infectious diseases (NIDs) between pre-pandemic (2010) and pandemic (2020). Similar studies in other regions also investigated the trend of NIDs from pre-pandemic to pandemic (Chen et al., 2024; Hirae, Hoshina & Koga, 2023; Li et al., 2024; Ullrich et al., 2021). Thus, we would consider this is the first study to investigate disease burden changes caused by NIDs from pre-pandemic to pandemic period. Further preventive measures and interventions could be addressed based on the specific diseases mentioned in this study with associated health administrations and policies.

Conclusion

Our results show that about 66% of the burden of disease is caused by death in the early stage of COVID-19 pandemic in 2020. The most important contributors to the burden of disease were tuberculosis, human immunodeficiency virus (HIV)/acquired immune deficiency syndrome (AIDS), and acute hepatitis B/D. Therefore, prevention and treatment plans should focus on these diseases. In addition, gender and age differences were observed in diseases spread by sexual contact and blood transmission, and air or droplet transmission. Therefore, interventions such as health education strategies should be implemented for these groups to reduce the incidence. An accurate estimation of the burden of infectious diseases can be used as a basis for public health policy formulation and to determine health planning priorities.

Supplemental Information

Supplemental Information 1 Supplementary information.

Supplemental Information 2 Raw data.

Additional Information and Declarations

Competing Interests

Author Contributions

Data Availability

The authors declare that they have no competing interests.

Ying-Fei Yang conceived and designed the experiments, performed the experiments, analyzed the data, prepared figures and/or tables, authored or reviewed drafts of the article, and approved the final draft.

Yu-Miao Chen conceived and designed the experiments, performed the experiments, analyzed the data, prepared figures and/or tables, authored or reviewed drafts of the article, and approved the final draft.

Si-Yu Chen performed the experiments, analyzed the data, authored or reviewed drafts of the article, and approved the final draft.

Po-Hao Chiu performed the experiments, analyzed the data, authored or reviewed drafts of the article, and approved the final draft.

Szu-Chieh Chen conceived and designed the experiments, performed the experiments, analyzed the data, prepared figures and/or tables, authored or reviewed drafts of the article, and approved the final draft.

The following information was supplied regarding data availability:

Data is available in the Supplemental File.

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
