# Peer review of "Burden changes in notifiable infectious diseases in Taiwan during the COVID-19 pandemic"

_PeerJ, doi:10.7717/peerj.18048_

## Round 0.1 · original submission · Major Revisions

· Academic Editor

Major Revisions

I thank the authors for submitting their very interesting manuscript to PeerJ. We have received the comments of two reviewers, comments that ought to improve the clarity of the manuscript for the eventual reader.

The comments of both reviewers will require attention but I will draw special attention to the comments of Reviewer #1, who has requested several clarifications but also some reflection on the use of population averages for DWs, as well as an explanation for the choice of using 2010 as the baseline for the analysis.

Reviewer 1 ·

Basic reporting

Verb tense problems & questionable word choices.
YLLs & YLDs acronyms are not defined in the main text.
Literature references are sufficient.

Experimental design

No rationale is provided as to why 2010 is used as the baseline. Why that year & why not multiple years?
The YLDs formula includes an I term that reflects age & sex, but it does not appear that the DWs or durations are based on age & sex. What is the point of the I term? Why not just multiply the DWs & durations by total number of cases?
The authors should provide reference(s) as to the validity of adding YLLs & YLDs to determine DALYs.
An important limitation is that DWs are not specific to age group. It's unclear to me that it is valid to use population averages given the very different impacts a disease can have depending on the patient's age.

Validity of the findings

I believe the y-axis title in Fig 1(e) should not include the word "of".
While I appreciate the attempt, the format of Fig 3A & 3B is unhelpful given how much larger the top 2 disease are relative to the others.
I suggest excluding the analyses in Fig 3C, 3D, & 3E & Fig 4C, 4D, & 4E given the exclusive focus elsewhere in the paper on annual data.
The results in Fig 1B vs 1D are suspicious. For example, why are the number of DC cases relatively similar in 2010 & 2020 but the DALYs so different? I understand why it is mathematically possible, but the authors should explore the data to assure the reader that the difference is accurate.

Additional comments

Please be more specific by what you mean by "parameters" in the 2nd sentence of the Materials & Methods section.

Reviewer 2 ·

Basic reporting

This study assessed the impact of COVID-19 on the burden of NID in Taiwan. Overall, the study demostrated that DAILY was reduced after NPI introduction for COVID-19. The study is well-designed and the manuscript is well-written.

Experimental design

The study design is well, and rigor. THe study method was described in detail.

Validity of the findings

Althought there is no novelty, the findings are confirmed.

Additional comments

I just have minor comments.
1. Please correct the term "non-communicable infectious diseases (NIDs)" in line 98.
2. Please add the timeline to show which NPI was implemented with time in 2020 in Taiwan.
3. English editing is needed.

---

## Round 0.2 · Minor Revisions

· Academic Editor

Minor Revisions

The reviewers have asked for some minor revisions to be done to the manuscript. Although there are many comments that have been returned, those comments are small and the suggested modifications should not take a long time. Most are requests for clarifications and additional explanations.

Please read all of the comments in full.

Reviewer 2 ·

Basic reporting

All the concerns have been well addressed by the authors. I have no more comment.

Experimental design

OK

Validity of the findings

Confirmed

Additional comments

No

Reviewer 3 ·

Basic reporting

The English is fluent and does not affect readability. However, it is recommended to recheck the verb tenses (past and present) in the article to ensure consistency. Literature references are sufficient.The research idea is good and the concept of study method is well-designed.

Experimental design

It is recommended to include the reasons for using 2010 as the research baseline directly in the article, as it will help readers understand better.

Additionally, although the COVID-19 pandemic outbreak occurred globally in 2020, Taiwan had relatively low confirmed cases in 2020 due to effective early policies. The real outbreak in Taiwan happened in 2021, with the highest number of daily confirmed cases in 2022 (https://www.worldometers.info/coronavirus/country/taiwan/). It is unclear why the study did not include data from 2020 to 2022, which could provide a clearer comparison.

The concept of disability weights (DW) is mentioned in the Materials and Methods section, but it is not further analyzed or explained in the Results and Discussion sections. It is only mentioned in the Limitations and Appendix. It is recommended to consider either removing the paragraph about DW from the Methods section and instead keeping it in the Limitations, mentioning the concept, or discussing it in the Discussion section with a reference to the Appendix.

Validity of the findings

In Fig 4A and 4B, the years 2020 and 2010 should be marked (similar to Figure 3) for clarity and ease of reading.
In Figures 3A, 3B, 4A, and 4B, the populations used are separately calculated for males and females. However, in the Results section of the article, the description is based on the total of males and females combined. It is recommended to include the total numbers (combined for males and females) in the figures to make them easier to understand.

The presentation of research results in Figures 3C-E and 4C-E is excellent, as the authors have included data from 2020 to 2022 for comparison, showing changes in the incidence rates of different diseases. However, there is minimal discussion explaining these findings in the Discussion section. It is recommended to delve deeper into the discussion and comparisons (similar to previous suggestions: since data is included up to 2022, why choose 2020 as a comparison year?), which can also correspond to the research goals mentioned in the Introduction section.

Additionally, it is recommended to add a concluding paragraph at the end of the discussion section to summarize the key points discussed. This will ensure that the discussion does not end abruptly and provides a cohesive closure to the section.

Additional comments

It is a commendable study. it is recommended to emphasize and highlight the research findings more prominently in the Discussion and Conclusion sections. The authors should express more about how these study results can inform public health policies.

Reviewer 4 ·

Basic reporting

.

Experimental design

.

Validity of the findings

.

Additional comments

General comment
The paper is not well organized. If the year and the burden in those years shown well as:
From Jan. to Dec. in 2010, 2016-2019 (pre-pandemic) and 2020-2022 (pandemic) will be good acceptable.

Language
Not self-written more copy, try to write what you want to show new in the finding

Title
Although it is possible to show a lot in the topic of the study, but the result does not show this, other similar studies have been done in the same topic, but this study has not been done in a different way and no gap filled.

Analysis
The data is collected from 2010 and 2020, but according to the topic, in order to see a sufficient change in the years, at least five years of resent data should be taken before 2020, otherwise it only shows a two-year comparison. A trend analysis should be done to show the effect and burden of this disease.

What is the difference with this study?
The impact of the coronavirus disease 2019 epidemic on notifiable infectious diseases in Taiwan: A database analysis (https://doi.org/10.1016/j.tmaid.2021.101997) out of adding the year 2010

Figure
The figures citation shows the data is taken from Jan. to Dec. in 2010, 2016-2019 (pre-pandemic) and 2020-2022 (pandemic), but not in the provided data set

Reference and citation
Use the standard format
(Chen et al., 2021; Lai et al., 2023; TCDC,
278 https://www.cdc.gov.tw/Category/ListContent/zKNFqsVWxUoUqE6fyhmBNA?uaid=L-
279 aa5EUo4UIJFsJlT7tAQw).
Line 545፣ line 550

---

## Round 0.3 · accepted · Accept

· Academic Editor

Accept

Thank you for the revisions to the manuscript, it is now ready for publication

Reviewer 2 ·

Basic reporting

The authors response well, so I have no more comment.

Experimental design

OK

Validity of the findings

OK

Additional comments

No

Reviewer 3 ·

Basic reporting

no comment

Experimental design

no comment

Validity of the findings

no comment

Additional comments

The revised content is thorough, making the overall argument of the article more complete.